# Clinical Determinants of Extraurinary Tract Recurrence and Survival after Radical Surgery for pT2 Upper Tract Urothelial Carcinoma

**DOI:** 10.3390/cancers15061858

**Published:** 2023-03-20

**Authors:** Yun-Ching Huang, Jui-Ming Liu, Hui-Ying Liu, Yin-Lun Chang, Chih-Shou Chen, Dong-Ru Ho, Chun-Te Wu, Miao-Fen Chen, Hung-Jen Wang, Hao-Lun Luo

**Affiliations:** 1Department of Urology, Chang Gung Memorial Hospital, Chiayi 613, Taiwan; dr5326@cgmh.org.tw (Y.-C.H.); cv7589@cgmh.org.tw (C.-S.C.); redox@cgmh.org.tw (D.-R.H.); 2Department of Medicine, College of Medicine, Chang Gung University, Taoyuan 333, Taiwan; 3Department of Urology, Taoyuan General Hospital, Ministry of Health and Welfare, Taoyuan 330, Taiwan; m5134@mail.tygh.gov.tw; 4Department of Obstetrics and Gynecology, Tri-Service General Hospital, National Defense Medical Center, Taipei 114, Taiwan; 5Department of Urology, Kaohsiung Chang Gung Memorial Hospital, Chang Gung University College of Medicine, Kaohsiung 833, Taiwan; ying1011@cgmh.org.tw (H.-Y.L.); tailanylyl@cgmh.org.tw (Y.-L.C.); 6Department of Urology, Chang Gung Memorial Hospital, Linkou, Taoyuan 333, Taiwan; chuntewu@cgmh.org.tw; 7Department of Radiation Oncology, Chang Gung Memorial Hospital, Linkou, Taoyuan 333, Taiwan; miaofen@cgmh.org.tw; 8Center for Shockwave Medicine and Tissue Engineering, Kaohsiung Chang Gung Memorial Hospital, Chang Gung University College of Medicine, Kaohsiung 833, Taiwan

**Keywords:** upper tract urothelial carcinoma, nephroureterectomy, ureterectomy, recurrence, survival

## Abstract

**Simple Summary:**

Although upper tract urothelial carcinoma (UTUC) is a relatively rare malignancy in Western countries, recurrence and distant metastasis are common even after definitive surgery. Many prognostic factors have been identified from previous studies, allowing clinicians to better stratify risk to select patients for perioperative systemic therapy; however, the applicability of adjuvant chemotherapy for patients with stage II UTUC after radical surgery remains unclear. In this study, we found that patients with primary tumor location at ureter or renal pelvis plus synchronous ureter had more frequent disease relapse and worse long-term oncological outcomes than other patients. Male sex, older age, history of previous bladder cancer, and positive surgical margins remain important unfavorable prognostic factors for recurrence and survival. Additional treatment and closer surveillance in patients with these negative prognostic factors are warranted despite complete pathological removal of the tumor.

**Abstract:**

Background: Oncologic outcomes for pT2N0M0 upper tract urothelial carcinoma (UTUC) after nephroureterectomy are not well defined, with most previous studies focused on a heterogeneous population. Therefore, we aimed to investigate the clinical determinants of extraurinary tract recurrence and survival after radical surgery in patients with localized UTUC. Methods: We retrospectively identified 476 patients with pT2N0M0 UTUC who underwent radical nephroureterectomy or ureterectomy between October 2002 and March 2022. To evaluate the prognostic impact, patients were divided into renal pelvic, ureteral, and both-region (renal pelvis plus synchronous ureter) groups based on tumor location. The outcomes included recurrence-free survival (RFS), cancer-specific survival (CSS), and overall survival (OS). Associations were evaluated using multivariable Cox regression analyses for prognostic factors and Kaplan–Meier analyses for survival curves. Results: The renal pelvic, ureteral, and both-region groups consisted of 151 (31.7%), 314 (66.0%), and 11 (2.3%) patients, respectively. Kaplan–Meier analyses comparing the three tumor types showed significant differences in 5-year RFS (83.6% vs. 73.6% vs. 52.5%, *p* = 0.013), CSS (88.6% vs. 80.7% vs. 51.0%, *p* = 0.011), and OS (83.4% vs. 70.1% vs. 45.6%, *p* = 0.002). Multivariable analyses showed that age >60 years, previous bladder cancer history, ureteral involvement (ureteral and both-regional groups), and positive surgical margins were significant negative prognostic factors for the studied outcomes. Conclusions: Patients with pT2 UTUC and presence of ureteral involvement had more frequent disease relapse. Subsequent adjuvant therapy regimens and close follow-up in patients with negative prognostic factors are warranted despite complete pathological removal of the tumor.

## 1. Introduction

Urothelial carcinoma is characterized by neoplastic growth of the entire urothelium, including the upper (renal pelvis and ureter) and the lower (bladder and urethra) urinary tract. Although upper tract urothelial carcinoma (UTUC) is a relatively uncommon malignancy in Western countries, making up only 5–10% of all urothelial carcinomas [1,2], it has a more advanced stage and worse differentiation than bladder cancer, as 62% of UTUCs are muscle-invasive at diagnosis compared to 35% of bladder cancers [3].

Locoregional control of non-muscle-invasive UTUC is satisfactory in definitive surgical series, with extraurinary tract recurrence and distant metastasis being rare [4]. However, in muscle-invasive UTUC, recurrence and distant metastasis are common even after radical nephroureterectomy [4]. A series from the UTUC collaboration showed 5-year recurrence-free survival (RFS) of 92%, 88%, 71%, 48%, and 5% for pTa/Tis, pT1, pT2, pT3, and pT4, respectively [5]. Therefore, adjuvant therapy should be considered for patients with muscle-invasive UTUC after definitive surgical therapy.

Many significant prognostic factors have been proposed based on previously published data [4], allowing clinicians to better stratify risk to select patients for subsequent adjuvant management; however, the use of adjuvant chemotherapy for patients with pT2 UTUC after radical surgery remains controversial. The 2022 National Comprehensive Cancer Network guidelines state that “adjuvant chemotherapy should be considered for patients with no platinum-based neoadjuvant treatment administered and pT3–4 or pN+ disease after surgery” [6]. Contrastingly, a phase 3, open-label, randomized controlled trial that enrolled 260 patients with pT2–4 or pN+ UTUC, of whom 74 (28%) had stage pT2 disease, concluded that adjuvant platinum-based chemotherapy after nephroureterectomy significantly improved disease-free survival [7].

Due to the lack of data on the utility of adjuvant therapy and population heterogeneity in previously published studies, prognostic predictors to identify patients with pT2 UTUC who are more likely to have extraurinary tract recurrence and should receive adjuvant chemotherapy and/or radiotherapy remain insufficient [4,8]. Therefore, this retrospective study aimed to evaluate the association between clinical characteristics and RFS, cancer-specific survival (CSS), and overall survival (OS) of the pT2 UTUC population and provide information to guide the postoperative management and prognostication of patients with pT2 UTUC after radical surgery. We hypothesized that the known risk factors for disease recurrence and survival after radical surgery would apply to the localized UTUC population.

## 2. Materials and Methods

### 2.1. Patient Selection

This study was performed with the approval and oversight of the Institutional Review Board (IRB No. 202100779B0). We retrospectively reviewed the medical charts of 476 consecutive patients with pT2 UTUC who were treated with radical nephroureterectomy or ureterectomy between October 2002 and March 2022 at three main branches of Chang Gung Memorial Hospital (Figure 1), which span northern to southern regions of Taiwan with high overall disease coverage [9]. Radical nephroureterectomy with bladder cuff excision is our institution’s standard treatment for patients with pT2 UTUC, with segmental ureteral resection performed in patients with distal ureteral tumors, serious renal insufficiency, or a solitary kidney. Patients with neoadjuvant chemotherapy, radiographic metastases, or retroperitoneal lymph node size > 1 cm were excluded. All patients underwent cystoscopy, chest radiography, and computed tomography (CT) urography or magnetic resonance urography (e.g., if any contraindications to CT urography were present) for preoperative risk stratification. In selected patients, diagnostic ureteroscopy, chest CT, and bone scan were used.

### 2.2. Pathological Evaluation

All the surgical specimens were examined by urologic pathologists at our institution. Tumors were staged according to the 2017 TNM classification by the American Joint Committee on Cancer for UTUC. Tumor grading was assessed according to the 2016 World Health Organization/International Society of Urological Pathology consensus classification. The pathological characteristics collected for predicting prognosis included tumor location, tumor grade, multifocal disease, carcinoma in situ (CIS), lymphovascular invasion (LVI), and surgical margin.

### 2.3. Outcome Measures

After surgery, patients were generally seen every three months for the first two years, every six months from the third through fifth year, and annually thereafter. Follow-up generally consisted of medical history, physical examination, blood laboratory tests, urinary cytology, renal ultrasound, and cystoscopic evaluation. Diagnostic imaging of both chest radiography and CT urography were used at least annually to detect locoregional recurrence and distant metastasis. Chest CT and bone scans were performed when clinically indicated.

The following clinical characteristics that may be associated with the outcomes were collected: sex, age at surgery, contralateral UTUC history (previous/synchronous/metachronous), bladder cancer history (previous/synchronous/metachronous), hydronephrosis grade, American Society of Anesthesiologists (ASA) score, diagnostic ureteroscopy (with/without biopsy), surgical approach (open/laparoscopic/robotic), surgical procedure (nephroureterectomy/ureterectomy), and estimated glomerular filtration rate (eGFR). Hydronephrosis grade was assessed by preoperative imaging, including CT, excretory urography, and renal ultrasonography. Hydronephrosis was reported as grade 0, no caliceal or pelvic dilatation; 1, pelvic dilatation only; 2, mild caliceal dilatation; 3, severe caliceal dilatation; and 4, caliceal dilatation accompanied by renal parenchymal atrophy [10]. eGFR was calculated using the 2021 Chronic Kidney Disease Epidemiology creatinine-based equation [11], which was the most widely used equation and recommended by the National Kidney Foundation and the American Society of Nephrology [12].

To evaluate the impact of clinical features on recurrence and survival, patients were divided into renal pelvic, ureteral, and both-region (renal pelvis plus synchronous ureter) groups based on the location of the muscle-invasive tumor at radical surgery (pT2).

Disease recurrence was defined as locoregional failure or distant metastases. Metachronous UC in the remnant genitourinary tract was not considered in the analysis of recurrence [13,14]. RFS interval was defined as the time between radical surgery and the first extraurinary tract recurrence, CSS interval was defined as the time between radical surgery and death from UTUC, and OS interval was defined as the time between radical surgery and death from any cause. Additionally, patients who died within 30 days of radical surgery or during hospital admission were censored at the time of death for the CSS and OS analysis [15].

### 2.4. Statistical Analysis

Continuous and categorical variables are presented as median values with interquartile ranges (IQR) and proportions, respectively. One-way ANOVA followed by the Tukey–Kramer test for post hoc comparisons [16] and chi-square test were used to compare continuous and categorical variables in the three groups, respectively. Survival curves were analyzed using the Kaplan–Meier method, and differences were determined using the log-rank test. The prognostic factors for RFS, CSS, and OS were estimated using the Cox proportional hazards regression model in the univariate and multivariate analyses. Only those factors with *p* < 0.05 in univariable analysis were further evaluated in multivariable analysis. All reported *p* values were two-sided, and statistical significance was set at *p* < 0.05. All statistical analyses were performed using SPSS version 20 (IBM Corp, Armonk, NY, USA) or Prism version 9 (GraphPad Software, San Diego, CA, USA).

## 3. Results

### 3.1. Baseline Characteristics

Patients’ clinical and pathological features stratified by tumor location are presented in Table 1. In the studied cohort, the renal pelvic, ureteral, and both-region groups comprised 151 (31.7%), 314 (66.0%), and 11 (2.3%) patients, respectively. The median age was 70.7 years (IQR, 62.4–77.0 years), and the proportion of female patients was 52.9% in the study population.

The proportion of patients with synchronous bladder cancer, metachronous bladder cancer, and extraurinary tract recurrence was significantly higher in the ureteral and both-region groups than in the renal pelvic group (*p* = 0.003, 0.004, and 0.025, respectively). Hydronephrosis, ureteroscopic biopsy, and ureterectomy were more commonly performed in the ureteral group than in the renal pelvic and both-region groups (*p* < 0.001, 0.001, and <0.001, respectively). Patients in the both-region group had a significantly higher proportion of multifocal disease than those in the renal pelvic and ureteral groups (*p* < 0.001). LVI was more common in the renal pelvic and both-region groups than in the ureteral group (*p* = 0.013). No statistically significant differences were observed in sex, age, contralateral UTUC history, previous bladder cancer history, ASA score, surgical approach, tumor grade, CIS, positive surgical margin, and eGFR between the groups (all, *p* > 0.05).

### 3.2. Recurrence and Survival

Median follow-up for the entire study cohort after surgery was 57.3 months (IQR, 24.1–100.2 months). At the end of the follow-up, 107 (21.0%) patients experienced extraurinary tract recurrence, 79 (18.9%) died of cancer-related causes, and 63 (12.6%) died of other causes. Of the 107 (21.0%) patients who had extraurinary tract recurrence, 42 (8.8%) had locoregional failure, 51 (10.7%) had distant metastasis, and 14 (2.9%) had locoregional failure plus synchronous distant metastasis, suggesting that the relatively common relapse pattern was distant metastasis (Table 1).

#### 3.2.1. Extraurinary Tract Recurrence

Extraurinary tract recurrence occurred in 23 (15.2%), 80 (25.5%), and 4 (36.4%) patients in the renal pelvic, ureteral, and both-region groups, respectively (*p* = 0.025). The median time interval of recurrence after radical surgery was 15.5 months (IQR, 7.4–32.9 months), and 73 (68.2%) patients with disease recurrence were identified within two years. The 5-year RFS was significantly higher in the renal pelvic group than in the ureteral and both-region groups (83.6% vs. 73.6% vs. 52.5%, *p* = 0.013; Figure 2).

In the multivariable analysis, previous bladder cancer history (*p* = 0.002), ureteral involvement, including ureteral and both-region groups (*p* = 0.022), and positive surgical margin (*p* < 0.001) were independent unfavorable prognostic factors for extraurinary tract recurrence (Table 2).

#### 3.2.2. Cancer-Specific Survival

Cancer-specific death occurred in 17 (11.3%), 58 (18.5%), and 4 (36.4%) patients in the renal pelvic, ureteral, and both-region groups, respectively (*p* = 0.030). The 5-year CSS was significantly higher in the renal pelvic group than in the ureteral and both-region groups (88.6% vs. 80.7% vs. 51.0%, *p* = 0.011; Figure 3). Multivariate analysis showed that male sex (*p* = 0.034), age > 60 years (*p* < 0.001), previous bladder cancer history (*p* = 0.001), ureteral involvement (*p* = 0.008), and positive surgical margins (*p* = 0.026) were significant negative prognostic factors for cancer-specific death (Table 3).

#### 3.2.3. Overall Survival

Death from any cause occurred in 33 (21.9%), 103 (32.8%), and 6 (54.5%) patients in the renal pelvic, ureteral, and both-region groups, respectively (*p* = 0.010). The 5-year OS was significantly higher in the renal pelvic group than in the ureteral and both-region groups (83.4% vs. 70.1% vs. 45.6%, *p* = 0.002; Figure 4). Age > 60 years (*p* < 0.001), previous bladder cancer history (*p* = 0.001), ureteral involvement (*p* = 0.005), and positive surgical margins (*p* = 0.009) were independently associated with OS in the multivariate Cox regression models (Table 4).

### 3.3. Subset Analysis

After excluding both-region cases, ureter tumor remained an unfavorable prognostic factor in the multivariate analysis for RFS (hazard ratio [HR] 1.85, 95% confidence interval [CI] 1.16–2.94; *p* = 0.010), CSS (HR 1.79, 95% CI 1.04–3.08; *p* = 0.035), and OS (HR 1.69, 95% CI 1.14–2.50; *p* = 0.009).

## 4. Discussion

Although many possible risk factors associated with recurrence and survival have been proposed [4], prognostic factors for patients with pT2 UTUC after radical surgery remain unclear. This may be due to the rare incidence of the disease and highly heterogeneous population enrolled in previous studies. Therefore, we only included patients with pT2N0M0 UTUC in this series. Although there are no data on the efficacy of postoperative chemotherapy or radiotherapy in terms of recurrence and mortality in patients with pT2N0M0 UTUC after radical surgery, our findings confirm the prognostic significance of several variables associated with disease recurrence and survival. Subsequent adjuvant therapy regimens and close follow-up in patients with poor prognostic factors are warranted despite complete pathological removal of the disease.

Previous studies have reported controversial results regarding the impact of primary tumor location on the outcome of UTUC treatment. Some studies failed to identify a difference in cancer-specific mortality between renal pelvic and ureteral tumors [17,18]. On the contrary, a retrospective study from multiple institutions, similar to our findings, showed a worse CSS for ureteral and both-region tumors than for renal pelvic tumors, even when adjusted for stage [19]. There are several possible explanations for the conflicting results between the present and previously published studies. In an international collaborative study from 13 centers worldwide, Raman et al. enrolled 1249 patients with UTUC managed by radical nephroureterectomy and assigned them into renal pelvic and ureteral groups [17]. After adjustment for pathologic tumor classification, grade, and lymph node status, tumor location did not independently predict cancer-specific mortality. Potential bias in this study may lie in the fact that tumors involving both the renal pelvis and ureteral regions were classified based on the dominant tumor location (in accordance with stage or size) under either the renal pelvic or ureteral group. In cases where renal pelvic and ureteral tumors are of the same stage, the tumor size is used to identify the tumor location. We believe this methodology can result in misclassification and bias, especially in an international retrospective study. Although the both-region group contained only 11 cases in this study, we postulate that tumors involving both the renal pelvis and ureter should be analyzed as distinct entities to avoid misclassification. In further analysis, tumor location remained a significant prognostic factor for RFS, CSS, and OS after excluding the both-region group. We also reported several important patient-related factors (e.g., history of previous bladder cancer, history of previous UTUC, and ASA score) and tumor-related factors (e.g., hydronephrosis, multifocality, CIS, lymphovascular invasion, and surgical margins) that were not assessed by Raman et al. [17]. In a similar report using administrative data from nine registries of the SEER database, Isbarn et al. identified 2824 patients treated with nephroureterectomy for UTUC and divided them into dichotomies according to primary tumor location [18]. Although the main findings were not different in terms of oncologic outcomes between patients with renal pelvic and ureteral tumors, data were collected by medical files review at participating institutions, thus introducing discrepancies in the interpretation of study variables. Overcoming these limitations, the study variables in the present study were reviewed by two independent urologists (YCH and HLL). Furthermore, this study was not a multi-institutional collaborative study, and practice patterns, including patients’ access to care, disease management, surgical techniques, and follow-up after surgery, were relatively uniform at our institute.

The poor prognosis of ureteral involvement can be explained in several ways. Compared to renal pelvic tumor with the natural barriers of renal parenchyma, perirenal fat, and Gerota’s fascia, ureteral involvement has a thin wall containing an extensive plexus of blood and lymphatic vessels, enabling easier invasion and spread of tumor cells [20]. Higher prevalence of hydronephrosis in ureteral involvement is also associated with more pronounced eGFR deterioration after radical nephroureterectomy [21], thereby restricting the use of postoperative cisplatin-based chemotherapy. Interestingly, a previous history of bladder cancer increased the risk of cancer-specific death by a 2.54-fold factor relative to no previous history of bladder cancer. Indeed, the proportion of multifocality and CIS was significantly higher in the group with a history of previous bladder cancer, which is a well-known predictor of poor outcomes after UTUC, than in the group with no previous history of bladder cancer, and this result is consistent with previous findings [22,23].

However, the effect of sex on the prognosis of UTUC after radical surgery remains unclear. Sikic et al. reported a 2.92-fold higher risk of cancer-specific death in female patients aged 59 years and older than in male patients [24]. Milojevic et al. found no significant difference in the CSS between female and male patients treated with radical nephroureterectomy [25]. In contrast, Wu et al. showed that male patients with UTUC were associated with more metachronous bladder cancer and higher cancer-specific mortality compared to female patients with UTUC [26]. Our observations that male patients with UTUC have worse cancer-specific mortality compared to female patients based on multivariate analysis are consistent with the previous findings. Multiple factors, including genetic background, environmental exposure, tumor biology, hormonal variation, and anatomical factors, may play a role in the reported sex differences. However, this finding is not in line with recently published data [27], and further epidemiologic and molecular research is required to address the impact of sex on the incidence, progression, and metastasis of patients with UTUC.

Unlike surgical margins, the effects of age on clinical outcomes in patients with UTUC have rarely been discussed. A large population-based study using the SEER database showed that older age is directly associated with a decrease in CSS after adjustment for stage, grade, and treatment type [28]. Reasons for this may be include changes in the biological potential of tumor cells, decreased host immunity with advancing age, or even different choices of treatment in elderly patients compared with younger patients [29]. In the current study, 3.2% of patients treated with segmental ureterectomy were aged < 60 years, and 13.0% were aged > 80 years, given the higher risk of residual disease [30]. Furthermore, elderly patients are less likely to undergo salvage chemotherapy for disease relapse [31], which has been shown to be associated with improved survival. These results indicate that treatment choice, at least in part, may account for the worse outcomes in older patients.

The current study had several limitations. First, due to the retrospective design of this study, biases are inevitable, as segmental ureteral resection in patients with distal ureteral tumors, with a serious renal insufficiency, or having a solitary kidney was chosen depending on patient preference after discussion with the treating urologist. Second, a centralized pathological review is lacking. The specimens being evaluated by various genitourinary pathologists over a long period could have led to discrepancies in the interpretation of the pathologic specimens. Third, the number of patients was too small to draw definite conclusions, particularly in the both-region group. Fourth, this study lacked data on adjuvant treatment. We could not confirm the effect of perioperative chemotherapy, immunotherapy, or radiotherapy on survival outcomes. Despite these limitations, our study was a relatively large cohort study focusing on the outcomes of pT2 UTUC after radical surgery. Our results indicate that patients with tumors located in the ureter or renal pelvis plus synchronous ureter could be candidates for additional treatment and closer follow-up after radical surgery. Prospective assessments to obtain a definitive role of adjuvant therapy in patients with pT2 UTUC are warranted.

## 5. Conclusions

Patients with pT2 UTUC and presence of ureteral involvement (ureteral and both-regional groups) had more frequent disease relapse and worse long-term oncological outcomes than other patients. Male sex, older age, history of previous bladder cancer, and positive surgical margins remain important unfavorable prognostic factors for recurrence and survival. Our findings support the need for more stringent follow-up strategies and subsequent adjuvant treatment in patients with those poor prognostic factors despite complete pathological removal of the disease. Prospectively large-scale studies investigating the role of tumor location in patients with pT2 UTUC are needed to obtain a definitive statement regarding this matter.

## Figures and Tables

**Figure 1 cancers-15-01858-f001:**
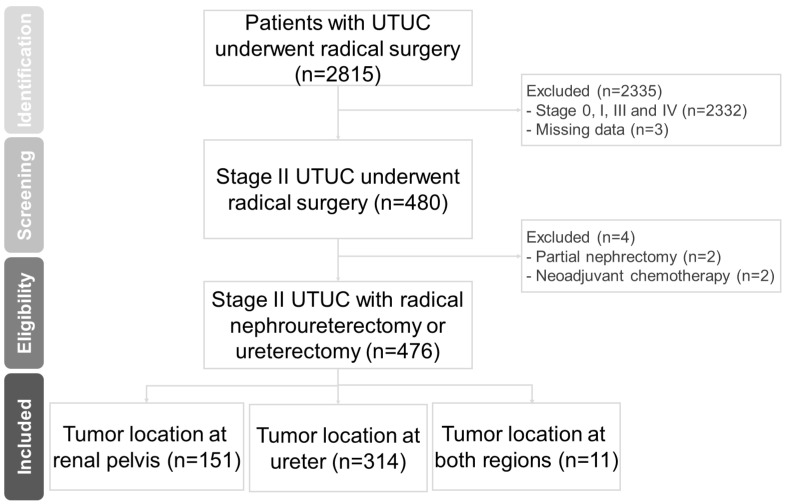
Flow chart for creation of the patient cohort dataset.

**Figure 2 cancers-15-01858-f002:**
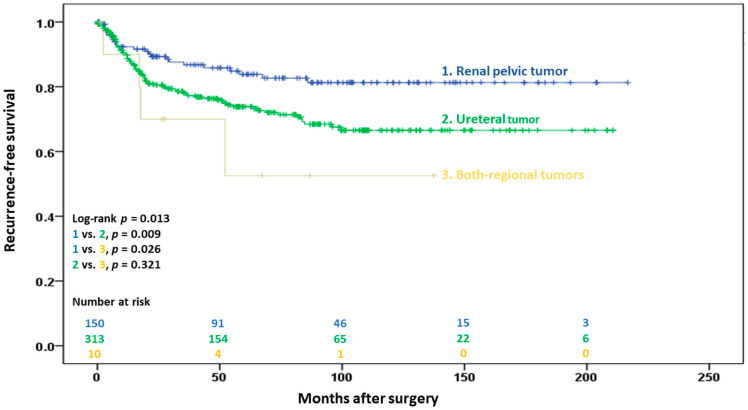
Kaplan–Meier estimates of extraurinary tract recurrence-free survival in 476 patients following radical surgery for pT2 upper urinary tract urothelial carcinoma, with stratification by tumor location.

**Figure 3 cancers-15-01858-f003:**
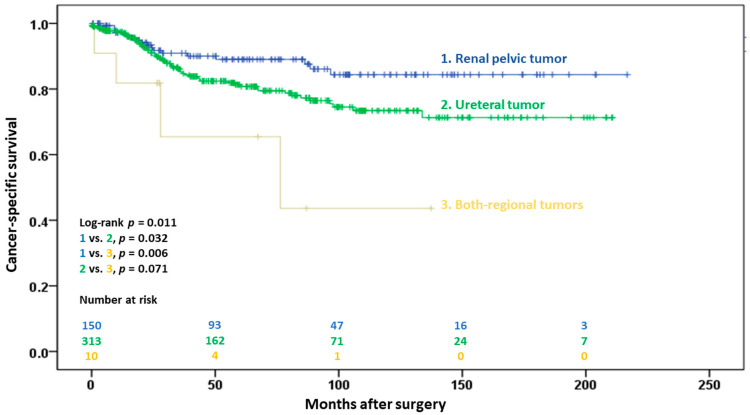
Kaplan–Meier estimates of cancer-specific survival in 476 patients following radical surgery for pT2 upper urinary tract urothelial carcinoma, with stratification by tumor location.

**Figure 4 cancers-15-01858-f004:**
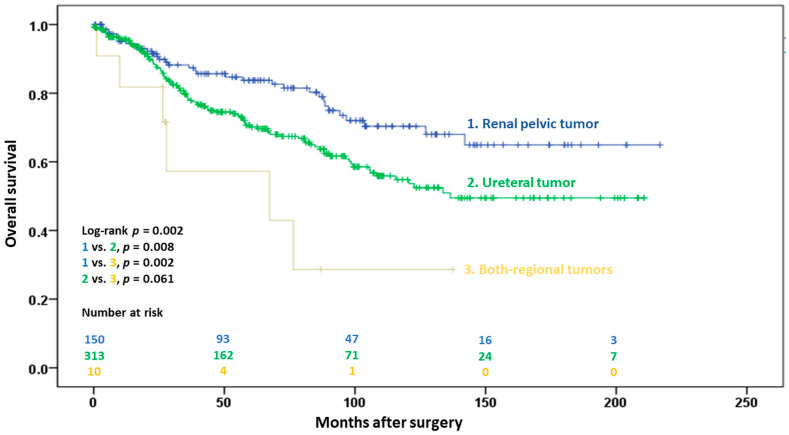
Kaplan–Meier estimates of overall survival in 476 patients following radical surgery for pT2 upper urinary tract urothelial carcinoma, with stratification by tumor location.

**Table 1 cancers-15-01858-t001:** Clinical and pathological characteristics.

		Main Tumor Location			
	Total (*n* = 476)	Renal Pelvis (*n* = 151)	Ureter (*n* = 314)	Both Regions (*n* = 11)	*p* Value
Gender					0.236
Female	252 (52.9)	87 (57.6)	161 (51.3)	4 (36.4)	
Male	224 (47.1)	64 (42.4)	153 (48.7)	7 (63.6)	
Age, years, median (IQR)	70.7 (62.4–77.0)	69.2 (61.2–76.1)	70.8 (62.7–77.5)	68.9 (58.5–74.0)	0.224
<60	94 (19.7)	34 (22.5)	56 (17.8)	4 (36.4)	0.439
60–70	133 (27.9)	44 (29.1)	87 (27.7)	2 (18.2)	
70–80	180 (37.8)	51 (33.8)	124 (39.5)	5 (45.5)	
>80	69 (14.5)	22 (14.6)	47 (15.0)	0 (0)	
Contralateral UTUC history					0.370
Previous	23 (4.8)	10 (6.6)	13 (4.1)	0 (0)	
Synchronous	7 (1.5)	2 (1.3)	4 (1.3)	1 (9.1)	
Metachronous	32 (6.7)	10 (6.6)	21 (6.7)	1 (9.1)	
Bladder cancer history					
Previous	58 (12.2)	16 (10.6)	39 (12.4)	3 (27.3)	0.258
Synchronous	93 (19.5)	18 (11.9)	70 (22.3)	5 (45.5)	0.003
Metachronous	158 (33.2)	37 (24.5)	114 (36.3)	7 (63.6)	0.004
Hydronephrosis grade					<0.001
0	41 (8.6)	25 (16.6)	15 (4.8)	1 (9.1)	
1	59 (12.4)	38 (25.2)	19 (6.1)	2 (18.2)	
2	114 (23.9)	39 (25.8)	71 (22.6)	4 (36.4)	
3	121 (25.4)	25 (16.6)	96 (30.6)	0 (0)	
4	130 (27.3)	21 (13.9)	105 (33.4)	4 (36.4)	
Unknown	11 (2.3)	3 (2.0)	8 (2.5)	0 (0)	
ASA score, median (IQR)	3 (2–3)	3 (2–3)	3 (2–3)	3 (2–3)	0.542
≤2	189 (39.7)	64 (42.4)	122 (38.9)	3 (27.3)	0.533
≥3	287 (60.3)	87 (57.6)	192 (61.1)	8 (72.7)	
Diagnostic ureteroscopy					0.001
Ureteroscopic biopsy	125 (26.3)	22 (14.6)	100 (31.8)	3 (27.3)	
Ureteroscopy without biopsy	100 (21.0)	31 (20.5)	68 (21.7)	1 (9.1)	
Surgical approach					0.608
Open	247 (51.9)	76 (50.3)	164 (52.2)	7 (63.6)	
Laparoscopic	213 (44.7)	72 (47.7)	137 (43.6)	4 (36.4)	
Robotic	16 (3.4)	3 (2.0)	13 (4.1)	0 (0)	
Surgical procedure					<0.001
Nephroureterectomy	440 (92.4)	151 (100.0)	278 (88.5)	11 (100.0)	
Ureterectomy	36 (7.6)	0 (0)	36 (11.5)	0 (0)	
Tumor grade					0.307
Low	25 (5.3)	11 (7.3)	13 (4.1)	1 (9.1)	
High	451 (94.7)	140 (92.7)	301 (95.9)	10 (90.9)	
Multifocal disease	141 (29.6)	38 (25.2)	92 (29.3)	11 (100.0)	<0.001
Carcinoma in situ	125 (26.3)	35 (23.2)	85 (27.1)	5 (45.5)	0.230
Lymphovascular invasion	54 (11.3)	26 (17.2)	26 (8.3)	2 (18.2)	0.013
Positive surgical margin	18 (3.8)	2 (1.3)	16 (5.1)	0 (0)	0.109
eGFR, mL/min/1.73 m^2^, median (IQR)	44.9 (24.3–57.0)	43.3 (20.8–56.8)	46.2 (27.2–57.1)	35.3 (0–53.9)	0.156
<60	362 (76.1)	117 (77.5)	235 (74.8)	10 (90.9)	0.315
≥60	97 (20.4)	26 (17.2)	70 (22.3)	1 (9.1)	
Unknown	17 (3.6)	8 (5.3)	9 (2.9)	0 (0)	
Recurrence	107 (22.5)	23 (15.2)	80 (25.5)	4 (36.4)	0.025
Locoregional failure	42 (8.8)	7 (4.6)	33 (10.5)	2 (18.2)	0.111
Distant metastasis	51 (10.7)	14 (9.3)	36 (11.5)	1 (9.1%)	
Locoregional + distant metastasis	14 (2.9)	2 (1.3)	11 (3.5)	1 (9.1)	

Data are *n* (%), unless otherwise stated. IQR: interquartile range, UTUC: upper urinary tract urothelial carcinoma, ASA: American Society of Anesthesiologists, eGFR: estimated glomerular filtration rate. Hydronephrosis grading scale, including grade 0—no caliceal or pelvic dilation, grade 1—pelvic dilatation only, grade 2—mild caliceal dilatation, grade 3—severe caliceal dilatation, and grade 4—renal parenchymal atrophy.

**Table 2 cancers-15-01858-t002:** Univariate and multivariate analysis predicting prognostic factors for recurrence-free survival in the patients with pT2 UTUC after radical surgery.

	Recurrence-Free Survival			
	Univariate		Multivariate	
	HR (95% CI)	*p* Value	HR (95% CI)	*p* Value
Male gender (referent: female)	1.54 (1.05–2.26)	0.026	1.41 (0.96–2.08)	0.080
Age (referent: <60 years)		0.085		0.056
60–70 years	1.84 (1.02–3.34)	0.044	2.12 (1.16–3.88)	0.015
70–80 years	1.70 (0.95–3.04)	0.073	1.86 (1.03–3.37)	0.040
>80	2.38 (1.20–4.74)	0.013	2.42 (1.21–4.87)	0.013
Contralateral UTUC (referent: absent)		0.766		
Previous	1.37 (0.64–2.96)	0.418		
Synchronous	0.69 (0.10–4.96)	0.713		
Metachronous	0.81 (0.38–1.75)	0.592		
Bladder cancer (referent: absent)				
Previous	2.20 (1.39–3.50)	0.001	2.12 (1.31–3.42)	0.002
Synchronous	1.28 (0.81–2.01)	0.289		
Metachronous	1.44 (0.98–2.11)	0.062		
Hydronephrosis grade (referent: grade 0)		0.209		
1	0.42 (0.16–1.11)	0.081		
2	0.98 (0.47–2.01)	0.945		
3	1.14 (0.56–2.32)	0.715		
4	0.88 (0.43–1.81)	0.725		
ASA score ≥ 3 (referent: ASA ≤ 2)	0.97 (0.67–1.43)	0.892		
Diagnostic ureteroscopy (referent: no)		0.453		
Ureteroscopic biopsy	1.27 (0.82–1.97)	0.282		
Ureteroscopy without biopsy	0.94 (0.57–1.54)	0.793		
Surgical approach (referent: open)		0.173		
Laparoscopic	1.11 (0.75–1.64)	0.610		
Robotic	2.24 (0.96–5.22)	0.062		
Ureterectomy procedure (referent: NU)	1.70 (0.93–3.10)	0.083		
Tumor location (referent: renal pelvis)		0.015		0.022
Ureter	1.85 (1.16–2.94)	0.010	1.77 (1.11–2.83)	0.016
Synchronous renal pelvis and ureter	3.09 (1.07–8.94)	0.038	3.18 (1.07–9.41)	0.037
Tumor grade (referent: low grade)	1.45 (0.53–3.94)	0.466		
Multifocal disease (referent: absent)	1.12 (0.75–1.69)	0.579		
Carcinoma in situ (referent: absent)	0.82 (0.52–1.28)	0.374		
Lymphovascular invasion (referent: absent)	1.50 (0.88–2.55)	0.135		
Positive surgical margin (referent: absent)	4.48 (2.33–8.61)	<0.001	3.79 (1.95–7.35)	<0.001
Chronic kidney disease ^a^ (referent: absent)	0.86 (0.54–1.35)	0.499		

Hydronephrosis grading scale, including grade 0—no caliceal or pelvic dilation, grade 1—pelvic dilatation only, grade 2—mild caliceal dilatation, grade 3—severe caliceal dilatation, and grade 4—renal parenchymal atrophy. ^a^ Chronic kidney disease was defined as estimated glomerular filtration rate less than 60 mL/min/1.73 m^2^. UTUC: upper urinary tract urothelial carcinoma; HR: hazard ratio; CI: confidence interval; ASA: American Society of Anesthesiologists; NU: nephroureterectomy.

**Table 3 cancers-15-01858-t003:** Univariate and multivariate analysis predicting prognostic factors for cancer-specific survival in the patients with pT2 UTUC after radical surgery.

	Cancer-Specific Survival			
	Univariate		Multivariate	
	HR (95% CI)	*p* Value	HR (95% CI)	*p* Value
Male gender (referent: female)	1.72 (1.10–2.69)	0.017	1.64 (1.04–2.59)	0.034
Age (referent: <60 years)		0.001		<0.001
60–70 years	3.87 (1.69–8.85)	0.001	4.89 (2.10–11.4)	<0.001
70–80 years	2.96 (1.29–6.78)	0.010	3.66 (1.57–8.52)	0.003
>80	5.86 (2.37–14.5)	<0.001	6.78 (2.69–17.1)	<0.001
Contralateral UTUC (referent: absent)		0.929		
Previous	1.35 (0.54–3.34)	0.523		
Synchronous	0.93 (0.13–6.67)	0.938		
Metachronous	1.10 (0.50–2.39)	0.817		
Bladder cancer (referent: absent)				
Previous	2.54 (1.50–4.32)	0.001	2.49 (1.44–4.32)	0.001
Synchronous	1.64 (1.0–2.71)	0.051		
Metachronous	1.30 (0.83–2.03)	0.249		
Hydronephrosis grade (referent: grade 0)		0.350		
1		0.555		
2	0.72 (0.24–2.14)	0.776		
3	0.87 (0.33–2.27)	0.474		
4	1.39 (0.57–3.39)	0.472		
ASA score ≥ 3 (referent: ASA ≤ 2)	1.41 (0.89–2.22)	0.146		
Diagnostic ureteroscopy (referent: no)		0.511		
Ureteroscopic biopsy	1.19 (0.71–1.98)	0.510		
Ureteroscopy without biopsy	0.81 (0.45–1.46)	0.483		
Surgical approach (referent: open)		0.365		
Laparoscopic	0.95 (0.60–1.49)	0.809		
Robotic	2.01 (0.72–5.60)	0.185		
Ureterectomy procedure (referent: NU)	1.10 (0.48–2.53)	0.823		
Tumor location (referent: renal pelvis)		0.015		0.008
Ureter	1.79 (1.04–3.08)	0.035	1.75 (1.01–3.02)	0.045
Synchronous renal pelvis and ureter	4.35 (1.46–13.0)	0.008	5.39 (1.76–16.5)	0.003
Tumor grade (referent: low grade)	1.36 (0.43–4.33)	0.598		
Multifocal disease (referent: absent)	1.34 (0.84–2.13)	0.221		
Carcinoma in situ (referent: absent)	0.97 (0.59–1.60)	0.971		
Lymphovascular invasion (referent: absent)	1.57 (0.85–2.90)	0.151		
Positive surgical margin (referent: absent)	3.48 (1.51–8.03)	0.004	2.64 (1.13–6.17)	0.026
Chronic kidney disease ^a^ (referent: absent)	0.81 (0.49–1.35)	0.424		

Hydronephrosis grading scale, including grade 0—no caliceal or pelvic dilation, grade 1—pelvic dilatation only, grade 2—mild caliceal dilatation, grade 3—severe caliceal dilatation, and grade 4—renal parenchymal atrophy. ^a^ Chronic kidney disease was defined as estimated glomerular filtration rate less than 60 mL/min/1.73 m^2^. UTUC: upper urinary tract urothelial carcinoma; HR: hazard ratio; CI: confidence interval; ASA: American Society of Anesthesiologists; NU: nephroureterectomy.

**Table 4 cancers-15-01858-t004:** Univariate and multivariate analysis predicting prognostic factors for overall survival in the patients with pT2 UTUC after radical surgery.

	Overall Survival			
	Univariate		Multivariate	
	HR (95% CI)	*p* Value	HR (95% CI)	*p* Value
Male gender (referent: female)	1.27 (0.91–1.76)	0.162		
Age (referent: <60 years)		<0.001		<0.001
60–70 years	2.38 (1.35–4.21)	0.003	2.63 (1.48–4.68)	0.001
70–80 years	2.88 (1.68–4.94)	<0.001	3.0 (1.72–5.23)	<0.001
>80	4.32 (2.29–8.16)	<0.001	4.04 (2.10–7.80)	<0.001
Contralateral UTUC (referent: absent)		0.860		
Previous	1.23 (0.60–2.52)	0.567		
Synchronous	1.47 (0.47–4.64)	0.507		
Metachronous	1.06 (0.60–1.89)	0.839		
Bladder cancer (referent: absent)				
Previous	2.24 (1.46–3.41)	<0.001	2.18 (1.40–3.40)	0.001
Synchronous	1.58 (1.08–2.31)	0.017	1.22 (0.81–1.83)	0.338
Metachronous	1.00 (0.71–1.41)	0.981		
Hydronephrosis grade (referent: grade 0)		0.280		
1	0.60 (0.27–1.33)	0.210		
2	1.02 (0.53–1.98)	0.954		
3	1.21 (0.64–2.30)	0.563		
4	1.17 (0.61–2.21)	0.640		
ASA score ≥ 3 (referent: ASA ≤ 2)	1.89 (1.33–2.70)	<0.001	1.40 (0.96–2.04)	0.084
Diagnostic ureteroscopy (referent: no)		0.301		
Ureteroscopic biopsy	1.16 (0.79–1.71)	0.447		
Ureteroscopy without biopsy	0.79 (0.51–1.23)	0.290		
Surgical approach (referent: open)		0.806		
Laparoscopic	0.91 (0.65–1.28)	0.596		
Robotic	1.17 (0.43–3.21)	0.757		
Ureterectomy procedure (referent: NU)	1.30 (0.72–2.35)	0.389		
Tumor location (referent: renal pelvis)		0.003		0.005
Ureter	1.69 (1.14–2.50)	0.009	1.62 (1.09–2.40)	0.018
Synchronous renal pelvis and ureter	3.67 (1.53–8.78)	0.003	3.84 (1.56–9.45)	0.003
Tumor grade (referent: low grade)	1.19 (0.52–2.69)	0.682		
Multifocal disease (referent: absent)	1.29 (0.91–1.83)	0.151		
Carcinoma in situ (referent: absent)	1.21 (0.85–1.72)	0.298		
Lymphovascular invasion (referent: absent)	1.33 (0.81–2.18)	0.263		
Positive surgical margin (referent: absent)	3.13 (1.59–6.17)	0.001	2.59 (1.27–5.25)	0.009
Chronic kidney disease ^a^ (referent: absent)	1.15 (0.76–1.75)	0.504		

Hydronephrosis grading scale, including grade 0—no caliceal or pelvic dilation, grade 1—pelvic dilatation only, grade 2—mild caliceal dilatation, grade 3—severe caliceal dilatation, and grade 4—renal parenchymal atrophy. ^a^ Chronic kidney disease was defined as estimated glomerular filtration rate less than 60 mL/min/1.73 m^2^. UTUC: upper urinary tract urothelial carcinoma; HR: hazard ratio; CI: confidence interval; ASA: American Society of Anesthesiologists; NU: nephroureterectomy.

## Data Availability

The datasets generated during and/or analyzed during the current study are available from the corresponding author on reasonable request.

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
