# Peer review of "Clinical Determinants of Extraurinary Tract Recurrence and Survival after Radical Surgery for pT2 Upper Tract Urothelial Carcinoma"

_cancers, 2023, doi:10.3390/cancers15061858_

Round 1
Reviewer 1 Report
Thank you for permitting me to review this manuscript
Introduction
Line 60 , why there is a reference to US only , other countries should also be listed if there is no reference , it should also be stated that the only data available are from US
Line 64 PPR (please provide reference)
figure 3 and figure 4 (are similar in shape) are the data to death related to other causes (non cancer ) available ?
conclusion :
please elaborate adjuvent treatement available for these particular case
In addition why authors suggest adjuvent therapy as they suggest in the introduction that there is no data on adjuvent therapy in the introduction , , this suggestion implies adding a miniparagraph in adjuvent therapy in the discussion section
In addition as authors recognise themselves that the presence of small number of double localized tumors caution is necessary in the conclusion related to this category of patients which necessitated future studies
Author Response
Thank you for permitting me to review this manuscript
Introduction
Line 60, why there is a reference to US only, other countries should also be listed if there is no reference, it should also be stated that the only data available are from US.
Ans: We appreciate the reviewer’s opinion and have included an additional reference related to the incidence of UTUC in the introduction section of our revised manuscript. Soria’s paper was cited as reference 2 in the revised manuscript.
Line 64 PPR (please provide reference)
Ans: Thanks for the reviewer’s suggestion. We have added the reference in the revised manuscript as reference 4.
figure 3 and figure 4 (are similar in shape) are the data to death related to other causes (no cancer) available?
Ans: Thanks again. Only 11 patients were enrolled in the both-region group. Therefore, the survival curve of cancer-specific survival (figure 3) and overall survival (figure 4) in the both-region group are similar. As mentioned in the results section, 79 patients died of cancer-related causes, and 63 died of other causes. Of whom death from other causes, 16, 45 and 2 patients occurred in the renal pelvic, ureteral and both-regional groups, respectively. Our purpose with this manuscript was to investigate the prognostic predictors of disease recurrence and death from UTUC. In order to keep the focus on the oncologic outcomes for UTUC, we did not provide additional information on non-UTUC-related death in the revised manuscript. Additionally, patients who died from any cause is showing in the figure 4.
conclusion:
please elaborate adjuvant treatment available for these particular case
Ans: Agree and revised according in the conclusion section.
In addition why authors suggest adjuvant therapy as they suggest in the introduction that there is no data on adjuvant therapy in the introduction, this suggestion implies adding a mini-paragraph in adjuvant therapy in the discussion section
Ans: We apologize for the lack of clarity in our prior submission. A series from the UTUC collaboration reported a high recurrent rate for pT2 UTUC after radical nephroureterectomy (see below reference 1). Thus, it is reasonable to consider adjuvant therapy in an effort to improve the survival rate. Due to the low incidence of UTUC, the efficacy of postoperative adjuvant therapies for patients with pT2N0M0 UTUC is not well defined. The European Association of Urology guidelines have been updated in 2023 on the EAU web site (http://uroweb.org) and state that post-operative platinum-based adjuvant chemotherapy improves cancer-specific survival in patients with pT2-4 or pN+ UTUC. Contrastingly, the 2022 National Comprehensive Cancer Network guidelines state that adjuvant chemotherapy should be considered for patients with no platinum-based neoadjuvant treatment administered and pT3-4 or pN+ disease after surgery (see below reference 2).
Due to lacked data on adjuvant treatment in the current study, our aim was to evaluate the association between clinical characteristics and survival, and provide information to guide the subsequent adjuvant management and prognostication of patients with pT2 UTUC after radical surgery. This information had been descripted in the section of introduction and discussion. Moreover, we certainly will incorporate the reviewer’s suggestions in our ongoing and future investigations.
References:
- Margulis V, et al. Outcomes of radical nephroureterectomy: a series from the Upper Tract Urothelial Carcinoma Collaboration. Cancer 2009, 115, 1224-1233.
- Flaig TW, et al. Bladder Cancer, Version 3.2020, NCCN Clinical Practice Guidelines in Oncology. J Natl Compr Canc Netw 2020, 18, 329-354.
In addition as authors recognize themselves that the presence of small number of double localized tumors caution is necessary in the conclusion related to this category of patients which necessitated future studies.
Ans: We’ve revised our conclusion section to be more circumspect in accordance with the reviewer’s recommendation. Thank you.
Reviewer 2 Report
The present study revealed the difference of survival outcomes among the locations of pT2 upper tract urothelial carcinoma (UTUC) after radical surgery in a Taiwan cohort. Moreover, the authors showed the prognostic factors of RFS, CSS and OS in multivariable analyses. These findings are important for urologists. However, there are some concerns to be addressed in the manuscript.
1. Why did the authors include variant histology? The results of the present study were affected, because the prognosis is heterogeneous.
2. Why is it difference in survival outcomes between pelvis and ureter in pT2 UTUC? Please explain the reasons in Discussion.
3. The authors should use “predictor” properly in the manuscript.
4. “Oncologic outcomes for pT2N0M0 upper tract urothelial carcinoma 34 (UTUC) after nephroureterectomy are rare” was described in Abstract. Please revise it, because it is a strange sentence.
5. Why did the authors perform multivariate analysis of RFS about ASA score in Table 2?
Author Response
The present study revealed the difference of survival outcomes among the locations of pT2 upper tract urothelial carcinoma (UTUC) after radical surgery in a Taiwan cohort. Moreover, the authors showed the prognostic factors of RFS, CSS and OS in multivariable analyses. These findings are important for urologists. However, there are some concerns to be addressed in the manuscript.
Why did the authors include variant histology? The results of the present study were affected, because the prognosis is heterogeneous.
Ans: We fully agree with the reviewer’s opinion. Variants of the urothelial carcinoma include squamous cell, glandular, sarcomatoid, micropapillary, neuroendocrine and lymphoepithelial morphology. All these variants are related to aggressiveness and poor clinical outcome. Nonetheless, there is evidence that the presence of these variants does not predict poor clinical outcome if the remaining clinicopathologic features are considered (see below reference 1 and 2). In order to diminish the effect of variant histology, we decided to delate this factor in the revised manuscript. Thanks again.
References:
- Rink M, et al. Impact of histological variants on clinical outcomes of patients with upper urinary tract urothelial carcinoma. J Urol. 2012 Aug;188(2):398-404.
- Xylinas E, et al. Impact of histological variants on oncological outcomes of patients with urothelial carcinoma of the bladder treated with radical cystectomy. Eur J Cancer. 2013 May;49(8):1889-97.
Why is it difference in survival outcomes between pelvis and ureter in pT2 UTUC? Please explain the reasons in Discussion.
Ans: We appreciate the reviewer raises an important point and extend the point in the discussion of revised manuscript. Thanks again.
The authors should use “predictor” properly in the manuscript.
Ans: Agree and revised according in the manuscript. Thank you.
“Oncologic outcomes for pT2N0M0 upper tract urothelial carcinoma (UTUC) after nephroureterectomy are rare” was described in Abstract. Please revise it, because it is a strange sentence.
Ans: Agree and revised according to the comment. Thank you.
Why did the authors perform multivariate analysis of RFS about ASA score in Table 2?
Ans: We apologize for the error in our prior submission and have revised in accordance with the reviewer’s recommendation. Only those factors with p < 0.05 in univariable analysis were further evaluated in multivariable analysis. Thanks again.
Round 2
Reviewer 2 Report
There are no additional comments.